# The Role of the Fatty Liver Index (FLI) in the Management of Non-Alcoholic Fatty Liver Disease: A Systematic Review

**DOI:** 10.3390/diagnostics13213316

**Published:** 2023-10-26

**Authors:** Teodora Biciusca, Sorina Ionelia Stan, Mara Amalia Balteanu, Ramona Cioboata, Alice Elena Ghenea, Suzana Danoiu, Ana-Maria Bumbea, Viorel Biciusca

**Affiliations:** 1Institute of Diagnostic and Interventional Radiology, Goethe University Hospital Frankfurt, 60596 Frankfurt am Main, Germany; teodora.biciusca@kgu.de; 2Doctoral School, University of Medicine and Pharmacy of Craiova, 200349 Craiova, Romania; sorina_stan@icloud.com; 3Department of Pneumology, Faculty of Medicine, Titu Maiorescu University, 031593 Bucharest, Romania; marapbalteanu@gmail.com; 4Department of Internal Medicine, Faculty of Medicine, University of Medicine and Pharmacy of Craiova, 200349 Craiova, Romania; biciuscaviorel@gmail.com; 5Department of Bacteriology-Virology-Parasitology, University of Medicine and Pharmacy of Craiova, 200349 Craiova, Romania; 6Department of Pathophysiology, Faculty of Medicine, University of Medicine and Pharmacy of Craiova, 200349 Craiova, Romania; suzanadanoiu@yahoo.com; 7Department of Medical Rehabilitation, Faculty of Medicine, University of Medicine and Pharmacy of Craiova, 200349 Craiova, Romania; anamariabumbea@yahoo.com

**Keywords:** fatty liver index, diagnostic, screening, non-alcoholic fatty liver disease

## Abstract

Currently, non-alcoholic fatty liver disease is the most common liver disease worldwide, with a prevalence of 32%. It is much more common among men (40%) and among patients with metabolic comorbidities such as obesity, diabetes and dyslipidemia. Being an asymptomatic disease, the diagnosis is often established on the basis of imaging methods, with an important role given to abdominal ultrasonography, computed tomography and magnetic resonance imaging. In order to facilitate diagnosis, experts have introduced a series of blood biomarkers. Two biomarker panels are currently validated for the diagnosis of non-alcoholic fatty liver disease: the fatty liver index, and the hepatic steatosis index. The fatty liver index has been in use in medical practice for over 17 years and has demonstrated its accuracy in various studies that compared it with other diagnostic methods, highlighted its role in screening patients with cardiovascular risk and validated the effects of different diets and drugs that are proposed for the treatment of the disease. In the management of non-alcoholic fatty liver disease, the fatty liver index is an important algorithm in the diagnosis and prognosis of patients with metabolic risk. Taking into account the diversity of drugs to be approved in the treatment of non-alcoholic fatty liver disease, the fatty liver index will become an effective tool in monitoring the effects of these therapies.

## 1. Introduction

Non-alcoholic fatty liver disease (NAFLD) is a chronic liver disease which is characterized by the accumulation of lipids in hepatocytes [1]. At present, it is the most common liver disease globally, with a prevalence of 32% among adults, and is more frequent in men (40%) than in women (26%) [2]. In medical practice, it is more common in patients with obesity, diabetes, dyslipidaemia or any metabolic disorder and is associated with an increased rate of death from cardiovascular causes [3]. NAFLD, especially evolving in non-alcoholic steatohepatitis (NASH), sometimes progresses to advanced fibrosis, liver cirrhosis and hepatocellular carcinoma (HCC) [4].

Because it is an asymptomatic disease, it is often recognized through imaging methods, such as abdominal ultrasonography (AU), and abdominal computed tomography (CT) [5]. Currently, in the case of obese/overweight, diabetic, and metabolic syndrome (MS) patients, the practical guidelines [6] recommend an initial clinical assessment that includes anthropometric measurements (height and weight to calculate body mass index (BMI)), vital signs, and laboratory tests [7]. After these investigations, the presence of hepatic steatosis (HS) is demonstrated by imaging techniques, blood markers/scores, or liver biopsy puncture (LBP) [8]. LBP is rarely used because it is an invasive procedure associated with various complications, including bleeding, pain, abdominal discomfort, and, rarely, death, which limits its practical utility [9]. 

Imagistic techniques, such as abdominal CT, magnetic resonance imaging (MRI), and liver ultrasonography, are commonly included in the diagnosis of NAFLD [10]. Among these techniques, liver ultrasonography is the most frequently used method in medical practice due to its simplicity and its non-invasive nature. AU can provide good diagnostic accuracy for detecting moderate-to-severe hepatic steatosis. Due to the limited capacity of CT to detect mild degrees of hepatic steatosis and to the potential for radiation exposure, CT can be regarded as an unsuitable imaging modality for the diagnosis of NAFLD [11]. 

To facilitate the diagnosis of the disease, experts have introduced a range of blood biomarkers and non-invasive algorithms into medical practice. Currently, only the fatty liver index (FLI) and the hepatic steatosis index (HSI) have been validated for diagnosing HS in epidemiologic studies or for identifying potential patients for further clinical and paraclinical investigation [12,13]. The two non-invasive scores [14,15], together with the SteatoTest [16] and the NAFLD-liver fat score (NAFLD-LFS) [17], have given satisfactory results that have been validated with ultrasound in the case of FLI, HSI, SteatoTest and with proton magnetic resonance spectroscopy (^1^H-MRS) for NAFLD-LFS. 

The FLI includes four clinical and biochemical variables: BMI, waist circumference (WC), triglycerides, and gamma-glutamyl transferase (GGT). In a study conducted by Bedogni et al. [14], a diagnosis of NAFLD was confirmed when the FLI value reached or exceeded 60, while a value below 30 excluded the diagnosis, demonstrating excellent diagnostic accuracy. Using ultrasound as a reference in 216 subjects with and 280 without suspected liver disease, the authors demonstrated that an FLI < 30 rules out the presence of steatosis with a sensitivity of 87% while an FLI ≥ 60 rules in the presence of steatosis with a specificity of 86% [14]. Confirmation of NAFLD by calculating the FLI has only been validated against liver ultrasonography [14].

The next step in the management of HS is to exclude other secondary causes, such as Wilson’s disease, hemochromatosis, autoimmune liver disease, alpha 1 antitrypsin deficiency, genetic disorders of lipid metabolism, chronic viral hepatitis C (especially genotype 3), chronic alcohol consumption and drug use which might induce steatosis (amiodarone, methotrexate, tamoxifen and corticosteroids) [18,19].

Another stage in the diagnosis of NAFLD is the assessment of liver fibrosis. In recent years, non-invasive methods have been proposed for the diagnosis of liver fibrosis [20]. Thus, the most well-known methods are liver elastography and blood markers of fibrosis. Elastography measures the stiffness of the tissues, and the assessment of liver fibrosis is based on ultrasound, which measures only a small component of the stiffness of the tissues [21,22,23].

Over the past 20 years, a number of non-invasive serum biomonitoring tools have been developed and we now have non-invasive tests for the assessment of liver fibrosis that have been validated against liver biopsy [24], such as the enhanced liver fibrosis test (ELFTM) [25], fibrosis-4 index (FIB-4) [26], NAFLD fibrosis score (NFS) [27], aspartate aminotransferase (AST)-to-platelet ratio index (APRI) [28] and FibroTest [29].

These non-invasive markers are used as a diagnostic assessment tool by which to detect patients who have advanced liver fibrosis and/or cirrhosis, offering an alternative and potential replacement to liver biopsy [30,31]. 

Along with the evolution of non-invasive tests for the evaluation and monitoring of HS and fibrosis, screening measures for NAFLD are increasingly important and are highlighted in international guidelines [32]. 

Currently, the most effective screening methods for NAFLD and fibrosis (FLI, HSI, Steatotest, FIB-4) are those based on serum biomarkers, as well as imaging methods such as transient elastography and MRI, although their cost is quite high [33].

In the most recent study, Zhang et al. [34] address this very current problem, discussing in detail the requirements of screening programs, their profitability, as well as the new recommendations of international guidelines. In addition, the authors describe in detail the main screening methods based on blood biomarkers and imaging, specifying both the advantages and disadvantages of these methods. 

The authors mention that these screening measures should be applied especially to patients with risk factors such as type 2 diabetes mellitus (T2DM), MS and to those who show increases in liver enzymes. 

Considering the global burden of NAFLD, there is a real need to improve awareness of the health risks of NAFLD. Thus, it is important to implement screening methods that include an assessment of multiple organs beyond liver damage [35].

Starting from the immediate need to validate as many non-invasive tools as possible for the diagnosis of NAFLD, in this article we seek to present the utility of FLI together with the results of the most representative studies, in order to demonstrate the benefits of the use of FLI in the diagnosis, screening and monitoring of steatosis in in patients with metabolic risk.

## 2. Material, Methods, and Results

This systematic review is organized in two parts. The first part presents an overview of the role of FLI in the diagnostic management of NAFLD and the second part presents an overview of the role of FLI in monitoring the therapeutic methods applied in HS. Each part was written while taking into account the PRISMA guidelines.

### 2.1. Part I: The Role of FLI in Diagnostic Management of NAFLD

#### 2.1.1. Search Literature

A literature search was performed using the following databases: Scopus, Medline and PubMed. The review included studies published in the last 21 years, between 1 June 2002 and 30 May 2023. In addition, a manual search was performed on the reference lists of the selected articles to increase the consulted literature.

#### 2.1.2. Selection Criteria for Studies

The selection of articles suitable to our research theme was carried out using themes and keywords such as: “non-alcoholic fatty liver disease”, “hepatic steatosis”, “non-invasive diagnostic methods”, “the role of FLI in the diagnosis and screening of hepatic steatosis”, and “the role of FLI in monitoring the therapeutic methods used in patients with NAFLD”.

##### Definition of NAFLD

There are several clinical forms of fatty liver. SLD defines hepatic steatosis of any cause. ALD defines hepatic steatosis due to alcohol. NAFLD defines non-alcoholic fatty liver disease. MASLD defines hepatic steatosis associated with metabolic dysfunction and includes non-alcoholic fatty liver associated with at least one metabolic risk factor (obesity or WC over 94 cm (males) or 80 cm (females); arterial hypertension over 130/85 mmHg or antihypertensive medication; fasting blood glucose over 126 mg/dL or values over 140 mg/dL at 2 h within impaired glucose tolerance (IGT) or glycated haemoglobin (HbA1c) over 5.7% or D2 diabetes or antidiabetic treatment; hypertriglyceridemia over 150 mmHg or specific lipid-lowering treatment; and low HDL-cholesterol values below 40 mg/dL (males) or below 50 mg/dL (females) [36]. In the evolution of NAFLD/MASLD, two stages are distinguished: (i) non-alcoholic fatty liver (NAFL), or steatotic liver associated with metabolic dysfunction (MAFL) which presents low liver morbidity; and (ii) non-alcoholic steatohepatitis (NASH) or steatohepatitis associated with metabolic dysfunction (MASH), which has a higher risk of progressive liver fibrosis and has substantial liver-related mortality. Importantly, both stages are associated with an increased risk of cardiovascular events and non-hepatic malignancies such as colorectal cancer. In patients with NAFLD-related cirrhosis, however, liver complications are the main cause of death [37,38].

##### Diagnosis and Screening of NAFLD

A diagnosis of NAFLD requires that there is evidence of HS on imaging or histology, and other causes of liver disease or steatosis have been excluded [39]. NAFLD screening is performed in people who are at risk of developing NAFLD, due to the presence of certain risk factors. Currently, NAFLD screening is required for patients with IR/T2DM, central obesity, dyslipidaemia, hypertension and MS. Most often, the screening is performed through biochemical tests and non-invasive tools.

##### FLI Score

FLI was calculated based on the formula: ey/(1 + ey) × 100, where: y = 0.953 × triglycerides (TGs) (mg/dL) + 0.139 × BMI (kg/m^2^) + 0.718 × GGT(U/L) + 0.053 × WC (cm) − 15,745. Values of FLI ranged from 0 to 100 [14].

#### 2.1.3. Search Strategy

The search strategy consisted of a combination of the following search themes related to Boolean terms:i.General search terms related to HS: NAFLD, HS and NASH;ii.Specific search terms related to the diagnostic methods of NAFLD: LBP, AU, CAP, MRI, CT, HSI and FibroTest.

#### 2.1.4. Selection of Articles

The selection process was carried out by two authors (I.S.S. and V.B.), based on the recommendations for conducting systematic reviews. These authors carried out the search strategy in all databases. The first step of the selection involved applying the inclusion criteria to select the articles with potentially relevant titles for the chosen topic. The summaries of the studies with titles considered relevant were taken for examination in the second stage. Whenever the abstracts did not meet the eligibility criteria or contained insufficient data to select a decision, both evaluators read the full text to assess the eligibility of the respective article. The two evaluators discussed among themselves the disagreements that arose at the end of the selection process. In addition, all articles were reviewed again based on the inclusion criteria by the principal investigator.

#### 2.1.5. Data Extraction

Data extraction was performed by two independent evaluators. Extracted data were documented on a Microsoft Excel 2010 (version 14.0) data extraction form. The data that were captured for the first objective were: details of the publication of the article (first author, year of publication), the FLI predictive accuracy, the sensitivity, the specificity, the optimal value of the diagnostic test, the reference methods by which the diagnosis of NAFLD was established, and the *p*-test value of the statistical method used to process these data. In order to allow the description of the studies, additional information was extracted about the study design and the value of the statistical test used to compare the diagnostic methods of FLI. The lead author acted as a data checker, assessing the accuracy of the data extracted from the included articles. Discrepancies in the extracted data identified by the main author were communicated to the two evaluators, and disagreements were resolved by mutual consensus.

#### 2.1.6. Results

Following the searches in the first stage, we identified 500 articles from the databases mentioned above. Later, we applied filters, such as review articles and prospective and retrospective studies and eliminated 150 articles. Of the remaining articles, 100 were selected from Scopus and Medline and 150 from PubMed. We subsequently removed 50 articles because they were duplicates. The screening process imposed the selection of the most relevant articles for our theme, which further imposed the elimination of 150 articles. Due to the lack of access to the summaries of certain articles and the fact that they did not coincide with our theme, we still eliminated 50 articles. Later, we applied the inclusion criteria mentioned above and further eliminated 59 articles. In the end, there were 22 articles that helped us present the role of FLI in the diagnosis of HS and 19 that analysed the screening role of FLI for NAFLD (Figure 1).

##### Description of Included Studies

The general characteristics of the selected studies that examined the role of FLI in the diagnosis of HS are included and displayed in Table 1. The included studies varied in type, encompassing retrospective, prospective, and experimental cohort studies, with most being cross-sectional studies. Among the included studies, 12 (54.54%) originated from Asia, while only 3 studies were conducted in the USA, and the remaining 7 studies (31.81%) were carried out in Europe. Of the 22 studies, 14 emphasized the diagnostic role of FLI in comparison with imaging methods, while 8 studies explored the diagnostic role of FLI in relation to other diagnostic scores. Most of the studies recognized the diagnostic significance of FLI (22 studies; 53.65%), whereas 19 studies (46.34%) focused on the role of FLI as a screening method.

##### Description of Studies That Appreciated the Diagnostic Role of FLI

In Table 1 we present the accuracy and limit value of FLI in the case of the most representative studies in the literature that appreciated the diagnostic role of FLI.

FLI was first described in 2006 by Bedogni et al. [14], who, starting from the finding that HS correlates with a series of clinical and paraclinical parameters, conducted a clinical study on 216 subjects with HS and 280 without HS. In this study he established the threshold values for excluding HS (cut-off < 30) and for confirmation of HS (cut-off > 60) diagnosed by AU.

Later, Balkau [40] conducted a prospective study that included 3811 patients, of which after 9 years of observation, 203 subjects had associated T2DM. The diagnosis of HS was established based on AU and FLI, correlated with the presence of diabetes (*p* < 0.0001). Taking these results into account, the authors concluded that FLI can be considered an efficient method of NAFLD diagnosis, but also an indirect marker of inadequate insulin secretion, requiring additional investigations.

Other studies have compared the accuracy of FLI with the accuracy of other non-invasive diagnostic markers (biochemical and imaging) that have already been compared with invasive diagnostic methods of NAFLD. Kahl S. et al. [43] concluded that there is a high agreement between FLI and SteatoTest and a moderate agreement between HSI and AU, and that, if the intermediate value of FLI is excluded, the diagnostic value compared with AU is high, making it easy to use in medical practice.

A Chinese study by Xia M-F [45] of 3548 participants in whom NAFLD was established by AU showed that MS, T2DM, fasting serum insulin, BMI, and AST/ aspartate aminotransferase (ALT) ratio were found to be independent predictors of NAFLD in the Chinese population. By including these predictive factors in an algorithm, the authors developed another non-invasive diagnostic score that exhibited an AUROC of 0.76 in a Chinese validation cohort and 0.73 in a Finnish cohort. The newly developed Chinese NAFLD score proved effective in identifying those who needed further screening for NAFLD.

Recently, a Taiwanese study evaluated the optimal threshold value of FLI for patient selection for NAFLD screening in the Taiwanese community using AU for the diagnosis of NAFLD. Following the selection of subjects, 746 were diagnosed with NAFLD and 625 represented the control patients. FLI correlated with the severity of fatty liver determined by AU, but also with the NFS, especially in women. Through its results, the study concludes that the cut-off value of FLI for the diagnosis of steatosis could be changed to 10 for women and 20 for men, in order to increase the accessibility of patients to AU [49].

Jeong S. et al. [61], to facilitate the identification of patients at increased metabolic risk for NAFLD, conducted a study using data from the Korean National Health and Nutrition Examination Survey (KNHANES) and developed a new diagnostic score, K-NAFLD, that includes female sex, WC, blood pressure systolic, fasting blood glucose, TGs and ALT. They showed that this score, together with FLI, can be used to estimate the metabolic risk in these patients.

In a recent review, Ahn S.B [62] investigated methods for HS assessment, noting limitations in the sole use of serum markers. Among the evaluated markers (FLI, NAFLD-LFS, HSI), the NAFLD-LFS had the best results, following external validation and had a diagnostic agreement with ultrasonography. Although new serum markers are emerging, their validation is still lacking. Detection of steatosis is crucial to prevent the progression of NASH and liver fibrosis, as they are usually asymptomatic. Serum biomarkers have advantages due to applicability, reproducibility and availability. A new non-invasive marker is needed in order to make an early diagnosis of steatosis, thus preventing the evolution towards steatohepatitis and then towards liver fibrosis.

Lars Lind et al. [48] conducted NAFLD screening in two groups of populations to which NAFLD was defined as liver fat exceeding 5.5% using MRI-derived proton density fat fraction (MRI-PDFF). The first group consisted of a population-based (310 subjects) sample of 50-year-old individuals and the second group was considered a high-risk group consisting of patients with a BMI > 25 kg/m^2^ and either high plasma TGs (≥1.7 mmol/L) or T2DM (310 subjects). To these people, NAFLD was defined as liver fat exceeding 5.5% using MRI-PDFF for accurate fat quantification. Different scoring systems (FLI, HSI, lipid accumulation product (LAP), and NAFLD-LFS) were evaluated using logistic regression models to predict NAFLD in both groups. The study revealed varying performance of the four NAFLD scores across the two groups. FLI showed the best performance in the population-based sample (NAFLD prevalence: 23%), while LFS performed better in the high-risk sample (NAFLD prevalence: 73%). These observed NAFLD prevalence rates aligned with other, similar, studies. 

In an attempt to validate non-invasive scores for the diagnosis of HS, Zhang et al. [51] set out to verify the accuracy of three diagnostic methods (FLI, LFS, and liver fat quantified in %) in comparison with MRI-PDFF, which represents the current gold standard for the diagnosis of NAFLD. The study enrolled 169 patients, of which 109 were diagnosed with NAFLD and 60 were control patients and demonstrated a sensitivity of 87% and a specificity of 58.5% for FLI. Additionally, the study demonstrated that the remaining diagnostic methods that were used can be accurate and could be used as diagnostic tools for patients with NAFLD.

Kim H.N. [58] conducted a study in which the effectiveness of Hounsfeld (HU) values obtained from liver CT scans was evaluated in the quantification and classification of liver fat content, especially in those with mild HS. The study included 142 patients who underwent liver biopsy or liver resection, of which 44 patients had an HS ≥ 5% and 98 patients had an HS < 5% or had no clinically significant HS. CT scan, HSI, and FLI demonstrated robust diagnostic performance, showing high sensitivity and specificity when detecting mild HS and showing double confirmation of non-invasive HS scores by CT scan and LBP or liver resection.

##### Description of Studies That Appreciated the Screening Role of FLI for NAFLD

Next, we analysed the studies in the specialized literature that researched the screening role of FLI for NAFLD in people and patients who presented risk factors for HS, such as age, sex, genetic factor, prediabetes, diabetes, obesity, arterial hypertension, cardiovascular diseases and MS. The general characteristics of these studies are included in Table 2.

A cross-sectional study by Sergio Fresneda et al. [70] examined the prevalence of NAFLD in Spanish working adults, focusing on gender and age differences. The study included 33.216 participants aged 18 to 65 years and found that NAFLD, as defined by FLI, was more common in men (27.9%) than women (6.8%) and increased with age. Men also showed poorer cardiovascular and anthropometric profiles. Both men and women with NAFLD were associated with factors such as age, HDL-cholesterol, social class, prediabetes, diabetes, prehypertension, hypertension, and smoking. Because the association between diabetes and hypertension with NAFLD was stronger in women, it is imperative that metabolic problems that may have a greater impact on liver health be preferentially investigated in women compared to men.

A review by Johanna K. DiStefano [71] emphasizes that NAFLD can affect lean individuals, and despite their better metabolic profiles, they face a similar risk of disease progression as obese individuals. Lean individuals with NAFLD may even experience more severe liver consequences and higher mortality rates. Because lean individuals are less likely to be screened for NAFLD, through FLI, due to the absence of early symptoms, the review calls for a clearer understanding of NAFLD’s natural history in this population and increased awareness of potential health risks. The review also discusses factors contributing to lean NAFLD, such as diet, genetics, menopausal status, and ethnicity. It highlights the need for greater inclusion of lean individuals in NAFLD-related clinical trials, improved screening methods, and tailored treatment strategies based on the underlying causes.

A study by Xiang Hu [72] found that individuals with a first-degree family history of diabetes (FHD) were more likely to have a higher FLI, indicating a greater risk of NAFLD. This increased risk was independent of age, gender, metabolic health, and smoking. In people with T2DM, NAFLD prevalence was significantly higher, and T2DM was associated with a faster progression of NAFLD to severe forms. FHD was associated with an increased FLI, suggesting a genetic predisposition to NAFLD. This risk was independent of glucose metabolism status, emphasizing the importance of early NAFLD screening and prevention in individuals with FHD, even in the absence of diabetes.

A study by Eugene Han et al. [73] examined changes in the prevalence of fatty liver disease in the Korean general population from 2009 to 2017. Using the FLI, the study found that the prevalence of fatty liver disease (FLI ≥ 60) in Korean adults aged 20 years and older increased from 13.3% in 2009 to 15.5% in 2017. This increase was particularly prominent among men and the younger age group (20–39). Furthermore, the study found that people with T2DM had the highest prevalence of fatty liver disease, with a steep increase seen in the young T2DM population. These findings highlight the increasing prevalence of fatty liver disease in Korea, with a focus on vulnerable groups such as young adults, men, and those with T2DM. In some categories of patients, NAFLD screening is required to use FLI.

Liver disease has become a significant cause of death in individuals with T2DM due to a common underlying factor, IR. T2DM is a major risk factor for NAFLD, characterized by liver fat accumulation. NAFLD can progress to NASH, leading to liver fibrosis, cirrhosis, HCC, and death. Liver fibrosis often goes unnoticed until advanced stages. Traditional diagnosis involves invasive LBP but non-invasive methods, such as serum biomarkers and imaging, are now available. Early detection of advanced liver fibrosis is crucial for timely intervention. A review by Alshaima Alhinai discusses non-invasive diagnostic tools for NAFLD and liver fibrosis in the context of T2DM, providing clinicians with practical approaches to manage this common comorbidity in diabetes care. This study discusses the importance of identifying advanced liver fibrosis in T2DM patients with NAFLD and the challenges in doing so. It highlights the way in which advanced NAFLD and T2DM pose burdens on the healthcare system and the need for effective screening methods. Various non-invasive tools and panels have been developed to diagnose NAFLD and fibrosis, but their exact role in T2DM patients is unclear. Imaging tools show promise, but the role of current biomarkers is uncertain. Future research should focus on optimizing diagnostic tools, validating newer tests, and developing combinations of tools for effective screening in T2DM patients [74].

Kitazawa’s exploration of the associations of FLI with incidental diabetes in the context of obesity and prediabetes reveals its utility as a screening tool. Their findings highlight the risk of developing diabetes associated with high FLI even in non-obese individuals with prediabetes. This study highlights the potential of FLI to identify high-risk individuals within apparently low-risk groups [75].

A study by María Arias-Fernández et al. [76] explores the prevalence of non-alcoholic fatty liver disease (NAFLD) in people with prediabetes and overweight/obesity and its association with cardiovascular risk factors. The analysis, based on data from an ongoing clinical trial, reveals a high overall prevalence of NAFLD (defined by a FLI ≥ 60) at 78%. Men in particular showed a more adverse cardiometabolic profile compared with women, with higher blood pressure, liver enzyme levels and cardiovascular risk. The study emphasizes the importance of addressing comorbidities related to cardiovascular risk in people with prediabetes, as well as the need to evaluate NAFLD through FLI in these categories of patients.

A meta-analysis by Marieke de Vries et al. [77] investigated the prevalence of NAFLD in patients with type 1 diabetes and examined associated characteristics and outcomes. The researchers conducted a comprehensive review of relevant studies published up to March 2020. Among the findings, the pooled prevalence of NAFLD by non-invasive methods in these patients was estimated to be 19.3%, with variations depending on the diagnostic method and NAFLD definition. Ultrasound studies reported a higher prevalence (27.1%) compared with MRI (8.6%), LBP (19.3%), or transient elastography (2.3%). These findings highlight the need for standardized NAFLD diagnosis in type 1 diabetes patients to better understand contributing factors and outcomes.

A study by Stefano Ciardullo et al. [78] assessed the prevalence of NAFLD and advanced fibrosis in AU adults categorized by blood pressure levels. The analysis, involving over 11,000 participants from the National Health and Nutrition Examination Survey (2005–2016), revealed that NAFLD prevalence increased with rising blood pressure levels, reaching 50.2% in individuals with elevated blood pressure. Moreover, patients with hypertension exhibited a higher prevalence of advanced fibrosis (3–9%). When applying screening guidelines to these patients, approximately 26.7% required referral to hepatologists, with a higher risk of referral seen in Hispanic individuals, those with diabetes mellitus, heart failure, and abnormal urinary albumin excretion. The study suggests that screening for NAFLD by non-invasive methods in hypertensive patients may be beneficial, although further research on cost-effectiveness is necessary.

Lee’s investigation into the impact of NAFLD on health outcomes reveals the crucial role of FLI. By performing health screenings at multiple time points, the authors establish a clear association between FLI and the risks of all-cause mortality, myocardial infarction (MI), and stroke. Those with consistently elevated FLI had increased mortality and cardiovascular risks, while favourable changes in FLI status affected mortality. This emphasizes the value of FLI in identifying individuals at high cardiovascular risk and emphasizes the clinical significance of NAFLD for the prevention and treatment of cardiovascular disease [79].

Chung’s cross-sectional study highlights the potential of FLI as a predictor of cardiovascular disease (CVD) risk. The authors establish a link between high FLI values and increased odds of a high Framingham CVD risk score, independent of confounding variables. This link between FLI and long-term cardiovascular events suggests a role for this non-invasive biomarker in CVD risk assessment in the general population, reinforcing the importance of addressing hepatic steatosis for cardiovascular health [80]. 

Part I of this study has allowed us to analyse the diagnostic and predictive values of the FLI score for HS, in comparison with other diagnostic methods of NAFLD.

### 2.2. Part II: The Role of FLI in Therapeutic Management of NAFLD

Part two was carried out with the aim of analysing the role of FLI in monitoring the effectiveness of various diets, lifestyle changes, bariatric interventions or therapeutic methods on NAFLD.

#### 2.2.1. Literature Search, Search Strategy and Eligibility Criteria

The electronic databases used for the literature search in Part I were used for Part II. Initially, we specifically searched for studies that had as their main aim the investigation of the main methods of influencing the evolution of NAFLD. In the specialized literature, we researched those studies in which the effects of the main methods of improving hepatic steatosis (diets, lifestyles, bariatric interventions, therapies) and the diagnostic methods by which these improvements were appreciated were analysed.

In cases where there were major adjustments according to the primary reviewer, such trials were excluded in the between-treatment testing procedure. The clinical trial search strategy consisted of a combination of the following search themes respectively, linked by the Boolean term AND:i.Disease-specific terms: NAFLD or HS or NASH;ii.Terms related to the influencing possibilities of NAFLD: diets or lifestyles, or bariatric interventions or HS therapies;iii.Terms related to other means of evaluation of HS: FLI or AU, CT or MRI, or FibroTest or CAP.

#### 2.2.2. Data Extraction

The selection process of the identified articles was conducted as described previously in part 1. All the data extracted were input to Microsoft Excel and given to two other independent assessors for further verification. The following data were extracted: publication details (first author, year of publication), title, AUROC, sensibility, specificity, diagnostic methods of NAFLD and *p* value.

#### 2.2.3. Synthesis of the Best Evidence: Levels of Evidence

The main author (TB) performed a synthesis of numerous studies on the role of FLI in monitoring various therapeutic methods. The best evidence synthesis rating was determined based on the number of studies that investigated the best diagnostic value of the FLI test. As there is currently no approved treatment for HS, lifestyle and diet modification are key elements in the therapeutic strategy of the disease. In this sense, many authors have researched the relationship between FLI, diet, lifestyle change and various therapeutic interventions that have the role of improving these factors.

#### 2.2.4. Results

##### Characteristics of Included Studies

The method by which to search for the most relevant articles when assessing the role of FLI in the therapeutic management of NAFLD is presented in Figure 2. Following the searches in the second stage, we identified 220 articles from the databases mentioned above. Later, we applied filters, such as prospective and retrospective study articles, and eliminated 130 articles. Of the remaining 90 articles, 40 were selected from Scopus, 40 from PubMed, and 10 from Medline. Ten articles were subsequently removed because they were duplicates. The screening process imposed the selection of the most relevant articles for our theme, which further imposed the elimination of 20 articles. In the next step, due to a lack of access to the summaries of certain articles, another 20 articles were further eliminated. Later, we applied the inclusion criteria mentioned above and further eliminated 20 articles. In the end, there were 20 articles that helped us present the role of FLI in monitoring the various therapeutic methods applied in NAFLD.

Of the 220 studies identified from the electronic databases, only 20 met the inclusion criteria. The majority of the studies did not meet the inclusion criteria because they did not present the discriminatory capacity of FLI in the prediction of NAFLD. 

The characteristics of the studies are presented in Table 3. Of the studies remaining in the second stage, 13 studies evaluated the role of FLI in diet monitoring, 2 studies evaluated the effects of lifestyle change, 1 study evaluated the role of bariatric surgery and 4 evaluated the role of FLI in monitoring the treatment.

##### The Role of FLI in Monitoring the Effects of Different Diets on NAFLD

One’s food diet, derived via food principles, can contribute to the appearance/accentuation of HS or to its improvement. Thus, diet plays a crucial role in the development of NAFLD [87].

Research by Cantoral, Kanerva, Cantero and Naomi examines the relationship between diet and FLI, shedding light on various dietary factors. A study by Cantoral et al. [84], published 4 years ago, aimed to evaluate the hepatic effects of high fructose consumption. The authors compared the intakes of different food sources of fructose in relation to two liver indices (FLI and HSI) that predict HS and in relation to the identification of NAFLD by MRI in young adults in Mexico. At the end of the study 100 of the enrolled patients (18%) presented with NAFLD, of which 44 patients were diagnosed with HS using HSI and 46 by FLI. When the authors compared the caloric intake of commercial foods and beverages that reported fructose as an ingredient on their labels with HS index values used to classify HS, they observed that foods and beverages with high fructose content were most often consumed by those who have developed FLI diagnosed steatosis (score ≥ 30), suggesting that fructose intake is associated with metabolic changes that increase the risk of NAFLD. Contrary to assumptions, Kanerva’s study disputes the direct link between high fructose intake and the prevalence of NAFLD through FLI [81]. In contrast, Cantero’s investigation demonstrates the potential of energy-restricted diets, such as the Metabolic Syndrome Reduction in Navarre (RESMENA) approach, to improve FLI and related markers. In particular, insoluble fibre and fruit intake emerged as key contributors [83].

Naomi’s study adds complexity by revealing a higher prevalence of FLI-defined NAFLD associated with increased consumption of sugar-sweetened and low/no-calorie beverages. Surprisingly, moderate consumption of fruit juice showed an inverse association, highlighting the nuanced interaction between beverages and FLI [88].

Many studies have analysed the favourable effects of an anti-inflammatory diet on HS. Thus, Mitra Darbandi’s study highlighted the role of the dietary inflammatory index (DII) in influencing FLI and liver health. Their findings indicate that a pro-inflammatory diet is associated with worse liver markers, while an anti-inflammatory diet could improve liver health, reduce obesity, and alleviate fatty liver [89].

Son et al. [90] investigated the impact of a calorie-restricted Korean-style Mediterranean diet (KMD) on metabolic parameters. The study showed significant reductions in FLI, indicating the potential of Mediterranean-style diets to improve liver health by addressing chronic inflammation and insulin resistance (IR).

Stefan Drinda and colleagues [91] explored the effects of intermittent fasting on FLI in individuals with and without T2DM. Their study demonstrated that intermittent fasting, accompanied by weight loss, resulted in a significant decrease in FLI. Furthermore, individuals with T2DM experienced a greater reduction in FLI after fasting. The study findings highlight the role of fasting in improving FLI and suggest its potential as a strategy to mitigate the risk of NAFLD. The observed improvements in FLI after fasting further emphasize the interconnection between metabolic health, weight loss, and liver function.

In their study involving overweight and obese women, Alessandro Leone and colleagues [92] revealed a noteworthy link between adherence to the Mediterranean diet and improved liver health. Analysing parameters such as FLI, NAFLD-FLS and HSI, they found that greater adherence to the Mediterranean diet was associated with lower values of these indices. This effect was particularly pronounced in premenopausal women with obesity.

Saman Khalatbari-Soltani [93] also showed that stronger adherence to the Mediterranean diet was linked to a lower prevalence of HS as measured by the FLI. However, this effect appears to be influenced by BMI, suggesting that the beneficial impact of the Mediterranean diet on liver health may be partially mediated by its ability to reduce adiposity. These results highlight the potential of the Mediterranean diet as a preventive measure against HS.

Chiara Gelli’s study focused on the synergistic effects of nutritional counselling on the Mediterranean diet and an active lifestyle in reducing the risk and severity of NAFLD. This study highlights the combined impact of dietary guidance and lifestyle changes in the effective management of NAFLD, with the Mediterranean diet playing a crucial role in this approach [94].

To demonstrate the benefits of a diet rich in vegetables and green fruits, as well as the advantages of carbohydrate restriction, De Nucci S et al. [86] conducted a study on 40 patients, in which they exchanged a portion of high-carbohydrate food with a portion of green leafy vegetables over a three month period and assessed liver and metabolic markers of NAFLD. Before and at the end of the study, the 24 of the 40 patients who were able to complete the study were evaluated for the diagnosis of NAFLD by FLI, FAST score and FibroScan. The authors found that after lifestyle changes, both scores for the diagnosis of steatosis improved, so they concluded that a simple and easily acceptable dietary change, such as changing a standard serving of vegetables to replace a portion of carbohydrates, improves NAFLD and metabolic markers, such as glycated haemoglobin and triglycerides, and facilitates weight loss.

Another Italian case-cohort study by Lampignano et al. [95] evaluated the effects of the Mediterranean diet (Med Diet) in a population of 1403 (53.6% male) participants, aged over 65 years, in whom the diagnosis of NAFLD was established using the FLI. Given the presence of steatosis at an FLI score > 60, the authors demonstrated that the Med Diet may help prevent HS.

##### The Role of FLI in Monitoring the Effects of Lifestyle Change on NAFLD

As there is no currently approved treatment, weight loss and lifestyle change are the mainstays of therapy for improving HS. A lot of authors have explored the relationship between lifestyle factors and FLI in patients with NAFLD.

Negin Kamari and colleagues [60]. explored the relationship between lifestyle factors and FLI in Iranian adults with NAFLD. The study highlighted the impact of lifestyle factors on the risk of NAFLD. Notably, high physical activity was associated with lower FLI in both men and women, emphasizing the importance of exercise in the management of NAFLD. The study findings reinforce the role of lifestyle change, including increased physical activity, in improving liver function and reducing the risk of NAFLD.

Christine Freer’s study examined whether combining progressive resistance training (PRT) with weight loss (WL) provides additional benefits in improving FLI for older adults with T2DM and NAFLD. The study concluded that PRT did not provide additional benefit over WL alone in improving FLI. This suggests that weight loss plays a critical role in influencing FLI in this population. While PRT has general health benefits, addressing metabolic risk factors, particularly weight loss, appears to be a key factor in the impact of FLI in individuals with T2DM and NAFLD [96].

##### The Role of FLI in Assessing the Effectiveness of Bariatric Interventions on NAFLD

Eduardo Espinet Collet’s study focused on bariatric endoscopy as a therapeutic option for NAFLD in obese patients. The study revealed substantial improvements in various parameters, including FLI, one year after the procedure. Their findings suggest that bariatric endoscopy can effectively reduce liver fat and improve liver function, positioning it as a safe and effective method for the management of obesity-related liver disease [97].

##### The Role of FLI in Assessing the Effectiveness of Drug Therapies on NAFLD

Although none of the international guidelines for the management of NAFLD have endorsed any drug substance, agents used to treat T2D have been shown to improve liver function tests in patients receiving this treatment.

Some years ago, a study of 55 Japanese patients with T2DM indicated that the administration of dapagliflozin (5 mg/day) or non-sodium glucose cotransporter 2 inhibitors (SGLT2i) for 6 months determined that the reduction in hepatic fat accumulation associated with a decrease in abdominal SFA can be assessed by the attenuation ratio of the liver to the spleen (L/S) using CT [98].

A study by Cho K.Y. [99] has pointed out that dapagliflozin leads to significant reductions in FLI, primarily driven by improvements in glycaemic control and insulin levels. These findings emphasize the relevance of glycaemic management in modulating FLI. While pioglitazone did not produce the same improvements in FLI, the study suggests that dapagliflozin may provide more favourable outcomes in patients with NAFLD due to its potential impact on glycaemic control and a subsequent reduction in liver fat content.

Beginning from such findings, Gastaldelli et al. [85] evaluated the efficacy of these treatments on liver steatosis and fibrosis in T2D patients. The authors evaluated the efficacy of once-weekly exenatide (EXE) plus once-daily dapagliflozin (DAPA) administration compared with each drug alone in reducing biomarkers of fatty liver/steatosis and fibrosis in a 104 week study in 695 T2D patients, not controlled by metformin monotherapy. They found that at week 28, biomarkers of fatty liver/steatosis and fibrosis (FLI and FIB-4) were reduced from baseline in all treatment groups. Additionally, FLI and FIB-4 score values decreased more in patients who received both drugs compared with those treated with monotherapy. The authors conclude that the combination of EXE plus DAPA showed stronger effects than EXE once weekly or DAPA daily in ameliorating markers of HS and fibrosis in T2D patients, but prospective studies are needed to validate these findings.

Given the favourable effects of SGLT2i luseogliflozin on liver dysfunction, Seino conducted a prospective study that included 55 patients with diabetes and liver dysfunction who received luseoglifazone for 52 weeks. The authors monitored liver functions by performing biochemical (AST, ALT, GGT, glycated hemoglobin-HbA1c, blood glucose, HOMA, ferritin) and immunological parameters (interleukin-6 (IL-6), and high-sensitivity C-reactive protein (hs-CRP)), but also clinical markers (BMI, WC, BP, FLI, FIB-4 and NFS) from the initial moment until weeks 12, 24 and 52. The results obtained show a significant decrease in the level of AST, ALT, GGT, FPG and HbA1c from baseline to week 52. Similar behaviour was obtained for body weight, BMI, WC and HOMA-IR, but HOMA-β was unchanged. The authors observed that FLI, ferritin, M2-BP and NAFLD fibrosis scores decreased significantly, while FIB-4 index and type IV collagen 7S domain did not change significantly. Unfavourable behaviour was also recorded for hs-CRP and IL-6 levels. These results demonstrate that luseoglifazone is beneficial in patients with T2DM and liver dysfunction, as it improves liver function without causing other significant changes in the liver in terms of inflammation and liver fibrosis [100].

The FLI is also used to assess the effectiveness of probiotic treatment on HS. In this regard, Nazarii Kobyliak and colleagues [101] conducted a double-blind randomized trial investigating the effects of a multi-strain probiotic called “Symbiter” on T2DM patients with NAFLD. The study demonstrated a significant reduction in FLI in the probiotic group, indicating a potential therapeutic benefit of probiotics in reducing liver fat content. This reduction in FLI was accompanied by decreased aminotransferase activity and decreased inflammatory markers, suggesting the ability of the probiotic to attenuate liver inflammation.

## 3. Discussions

The identification of non-invasive methods for the diagnosis of NAFLD is fundamental, as these methods can aid in monitoring treatment response and tracking the progression of the disease [102]. On one hand, the diagnostic and prognostic performance of these non-invasive tools has been limited, hindering their clinical application for NAFLD screening and fibrosis staging. On the other hand, meta-analyses have shown that combining multiple serum biomarkers into panels can enhance their diagnostic performance. Therefore, this combined approach holds promise for the future application of non-invasive scores in diagnosing and staging steatosis [103]. One such non-invasive approach is FLI, which has proven its efficiency and accuracy in various studies related to NAFLD management. It is used for diagnosing, screening, and monitoring NAFLD in the context of diets, bariatric interventions, and treatments [104].

In all of the studies we examined, FLI’s diagnostic accuracy was consistently confirmed by another validated diagnostic method, often involving liver biopsy and/or imaging methods. LBP, the gold standard for diagnosing hepatic steatosis (HS), involves analysing a liver tissue fragment [105]. This test should be considered in all patients with persistently elevated aminotransferases when the diagnosis of NAFLD remains uncertain [106]. NASH diagnosis requires histological evidence of steatosis, lobular inflammation, and hepatocyte bloat with or without perisinusoidal fibrosis [107]. However, LBP is not suitable for all patients due to its invasiveness, cost, and associated risks [108]. Fedchuk et al. compared FLI results with liver biopsy results and found that FLI had a good agreement (AUC = 0.83) for diagnosing steatosis [42].

AU has been widely used to noninvasively evaluate HS and has also been compared with FLI values [109]. It indirectly assesses liver fat content based on subjective qualitative sonographic features such as liver echogenicity, echotexture, vessel visibility, and beam attenuation [110]. Since Bedogni’s (2006) [14] study, the diagnosis of HS has been confirmed by AU. Threshold values have been established, with FLI < 30 ruling out steatosis (sensitivity = 87%) and FLI ≥ 60 confirming it (specificity = 86%). The utility of AU in the objective quantification of steatosis is hampered by factors such as fibrosis, edema, and extrahepatic adipose tissue, which may lead to unavoidable errors in fat quantification [111].

Fibroscan elastography or controlled attenuation parameter (CAP) is an ultrasound-based test that simultaneously evaluates steatosis and liver fibrosis [112]. Transient elastography measures the attenuation of the AU beam as it traverses liver tissue, which grows in fatty liver, and is obtained by analyzing the ultrasonic signal acquired by the transient elastography device [113]. It is considered to be an accurate tool for the diagnosis and staging of HS with average area under the receiver operating characteristic (AUROC) values of 0.9 for mild, 0.8 for moderate, and 0.7 for severe steatosis [114]. Fibroscan can also assess liver fibrosis, which, once present, can affect the liver’s functional capacity by reducing blood flow and leading to serious conditions such as cirrhosis, liver cancer and liver failure. The AUROCs for AU and CAP in detecting advanced fibrosis were 58.2% and 82.3%, respectively, and for patients without fibrosis, they were 86.4% and 68.6 [115].

Unenhanced CT is a widely used imaging modality that indirectly evaluates HS based on liver X-ray attenuation [116]. The normal liver has an attenuation value of 50–65 Hounsfield units (HU), which is generally 8–10 HU higher than that of the spleen [117]. On non-contrast CT, normal liver parenchyma is approximately 50 to 60 HU, while fat is −20 to −100 HU. Due to the inconsistency in HU calibration by external factors, “fat-free spleen” can be used as an internal reference [118]. Hepatic HU < 40 has been suggested as a cut-off value for steatosis (>30%) [119]. Therefore, CT is a non-invasive imaging method frequently used to confirm the diagnosis of NAFLD in patients in whom HSI or FLI has been used as a diagnostic marker for HS. Kim’s study reported AUROCs for mild hepatic steatosis of 0.810 for HSI, 0.732 for liver HU value, 0.802 for the difference between liver and spleen HU value (L-S HU value), and 0.813 for FLI [58].

MRI, although expensive and less practical, offers high sensitivity and specificity for detecting HS, particularly MRI-PDFF, which outperforms CAP. In a study, Park et al. used LBP to compare the accuracy of MRI-PDFF with the accuracy of CAP in determining both hepatic steatosis and fibrosis in a group of 104 patients who were diagnosed with NAFLD. The study reported an AUROC value for MRI-PDFF of 0, 99 (95% CI, 0.98–1.00), which was higher in comparison with the CAP value (AUROC 0.85; 95% CI 0.75–0.96). Unfortunately, MRI is not an easy-to-use tool in medical practice due to the long scan time, its logistical complexity and the lack of expertise in the imaging centers [120]. Another accurate MRI technique, 1H magnetic resonance spectroscopy (1H-MRS), is rarely used due to its cost and complexity. In a recent meta-analysis, Jia and colleagues found that MRI-PDFF and CAP are highly accurate in quantifying steatosis in children and adolescents, with MRI-PDFF showing better results. MRI-PDFF had a sensitivity of 0.95 and specificity of 0.92 for diagnosing S1–S3 steatosis, much better results compared with CAP. This suggests that both methods are highly accurate in detecting steatosis, though MRI-PDFF grades steatosis a lot more significantly [121].

The other three scores currently used in the non-invasive diagnosis of HS, which were compared with the effectiveness of the FLI, were HSI, NAFLD score and SteatoTest. HSI, a user-friendly tool in medical practice, calculates the use of parameters such as AST/ALT ratio, BMI, gender (especially female), and the presence of type 2 diabetes (T2DM). It is validated for NAFLD screening, with a demonstrated accuracy indicated by an AUROC of 0.81 (95% CI 0.800–0.836). A value < 30 excludes steatosis while a value >36 confirms the presence of steatosis with a sensitivity of 93% [122]. SteatoTest is another diagnostic algorithm for HS and the only one that uses the biopsy as a reference. This diagnostic algorithm includes 12 parameters, including sex, age, BMI, ALT, apolipoprotein A-1, alpha2-macroglobulin, haptoglobin, total bilirubin, GGT, triglycerides, cholesterol and blood glucose, and with values ranging from 0 to 1. Its AUROC value for diagnosing HS is 0.80 with a sensitivity of 90% and a specificity of 88% [123]. The NAFLD score, validated in 2009 by Kotronen [15], incorporates parameters such as T2DM, MS, fasting serum insulin, AST and AST/ALT ratio. It boasts an AUROC value of 0.87 and the highest sensitivity and specificity of 95% compared with other scores. However, its practical utility is limited by the FLI since it also needs to estimate metabolic risks, such as obesity, insulin resistance and MS. The reference values to indicate or exclude steatosis are as follows: a score < −0.64 excludes steatosis, a score > −0.64 to <0.16 suggests steatosis affecting 5–33% of hepatocytes, and a score > 0.16 indicates steatosis affecting between 33–66% between hepatocytes [124].

Numerous investigations have explored how NAFLD affects overall health and have highlighted the significant role of FLI in early fibrosis detection. These studies offer valuable models for utilizing FLI as a straightforward predictor in medical practice, particularly for assessing cardiovascular risk in individuals with obesity, prediabetes, and type 2 diabetes (T2DM) [79,80,125]. FLI has emerged as a valuable marker for assessing the severity of NAFLD and predicting associated metabolic risks.

Multiple studies, conducted by various researchers, have delved into the impact of diets and different therapies on FLI, shedding light on potential strategies for managing and mitigating NAFLD. These studies have facilitated the understanding of how certain dietary factors influence the risk of NAFLD and FLI. The studies that investigated the effects of diets on hepatic steatosis and FLI revealed the positive effects of dietary intervention on body weight, metabolic parameters, and FLI [81,83,88,97]. One study has suggested that higher FLI values in individuals with elevated ALT levels correspond to greater improvements in liver parameters. This study reinforces FLI’s role in evaluating treatment outcomes for liver health [126]. The Mediterranean diet, known more as an eating pattern than a structured diet, gained popularity in the 1990s and has shown favourable effects on both steatosis and the FLI score [90,91,92,93,94]. In addition to benefiting individuals with NAFLD, the Mediterranean diet has been associated with a decreased risk of heart disease, lower blood pressure and improved LDL cholesterol. This is significant because heart disease and diabetes are strongly associated risk factors for fatty liver disease [127].

Apart from adopting a low-calorie diet to improve lifestyle, physical exercise has also demonstrated a crucial role in reducing liver fat content and, consequently, FLI. Research consistently supports the importance of lifestyle changes, including increased physical activity, in improving liver function and reducing the risk of NAFLD [60,96].

Studies examining the effects of bariatric surgery on hepatic steatosis have shown significant improvements in FLI after bariatric surgery, indicating a trend toward reduced liver fat and improvement in hepatic steatosis. This highlights the potential of bariatric surgery not only in addressing obesity but also in positively influencing NAFLD-related indices [128].

Studies that examined the effects of antidiabetics on HS have revealed that administration of dapagliflozin leads to significant reductions in FLI, improving glycaemic control and insulin levels. These findings emphasize the relevance of glycaemic management in modulating FLI [98,99]. The same studies suggest that dapagliflozin may provide more favorable outcomes in patients with NAFLD due to its potential impact on glycaemic control and the subsequent reduction in liver fat content. Regarding pioglitazone, it did not produce the same improvements in FLI as dapagliflozin [98]. Regarding the effects of probiotics on hepatic steatosis, research by Kobyliak N. et al. has demonstrated that the “Symbiter” probiotic holds promise in NAFLD therapy by improving liver fat content and reducing inflammation [101].

Collectively, the multitude of studies focused on dietary and lifestyle modifications, as well as various therapeutic methods emphasize the crucial role of different therapies and diets in modulating FLI and addressing NAFLD. Dietary modifications, bariatric interventions, probiotics, and drugs have all demonstrated the potential to impact hepatic steatosis and FLI, either directly by reducing liver fat content or indirectly by improving metabolic parameters. These insights emphasize the importance of multifaceted approaches in the management of NAFLD, paving the way for personalized interventions tailored to individual patient needs and the synergy between different therapeutic strategies.

## 4. Study Strengths and Limitations

The present study focuses on the utility of the FLI diagnostic score in both diagnosing and screening steatosis among patients with metabolic risk factors, particularly in evaluating the efficacy of specific treatment approaches for hepatic steatosis. Notably, this study distinguishes itself by conducting a more comprehensive analysis compared with previous research efforts, as it assesses and compares both invasive and non-invasive diagnostic methods for hepatic steatosis. In recent years, the introduction and utilization of novel antidiabetic and hypolipidemic medications in the management of hepatic steatosis have underscored the necessity of identifying non-invasive, cost-effective, and replicable techniques for monitoring the impact of these therapies on steatosis.

Nonetheless, it is essential to acknowledge certain limitations. Firstly, the study refrained from employing liver biopsy, widely regarded as the gold standard for hepatic steatosis diagnosis, due to the invasiveness and associated risks faced by patients. Secondly, the study protocol was not registered in the public PROSPERO platform, primarily due to extended waiting periods for registration.

## 5. Conclusions

In the management of NAFLD, FLI is an important algorithm in the diagnosis and prognosis of patients with metabolic risk. Considering the great diversity of drugs to be approved in the treatment of NAFLD, FLI will become an easy tool to use in evaluating the benefits of these therapies.

## Figures and Tables

**Figure 1 diagnostics-13-03316-f001:**
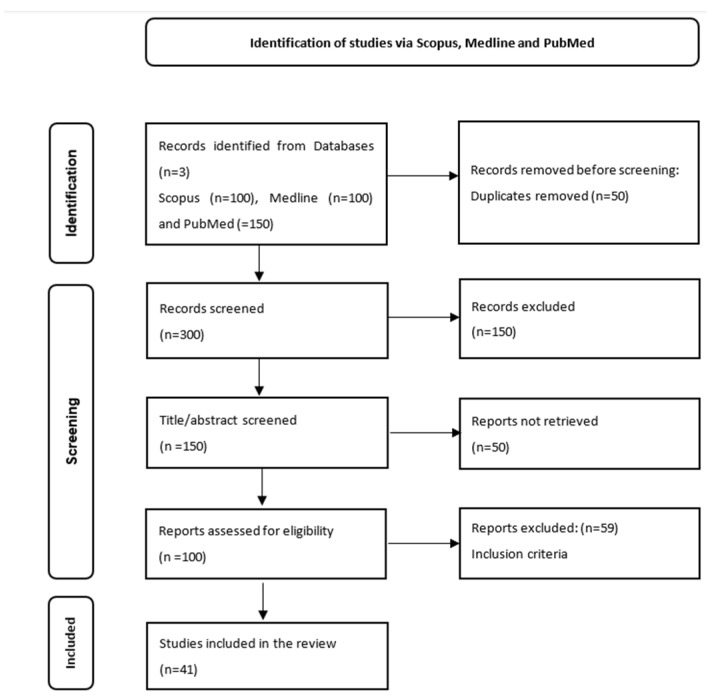
Prisma flow chart for the identification of articles related to the role of FLI in the diagnosis and screening of HS.

**Figure 2 diagnostics-13-03316-f002:**
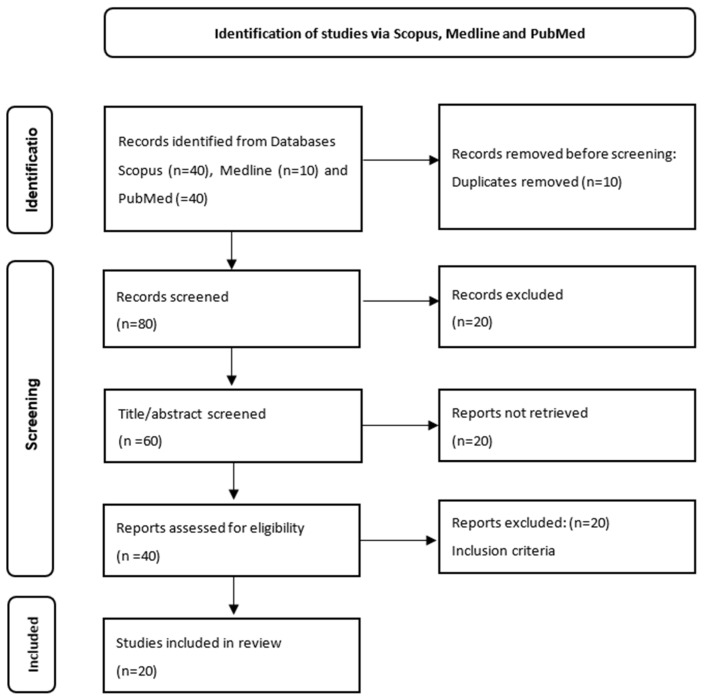
PRISMA diagram for identifying articles that evaluated the role of FLI in NAFLD screening.

**Table 1 diagnostics-13-03316-t001:** The accuracy and limit values of FLI in NAFLD diagnosis.

Author [Ref], Year	AUC (95% CI) FLI	Sensitivity % FLI vs. Other Diagnostic Methods	Specificity % FLI vs. Other Diagnostic Methods	Cut-Off FLI	Reference Methods for NAFLD Diagnosis	* *p* Value
Bedogni et al. [14] 2006	0.84 (0.81–0.87)	87 61	44 86	<30 ≥60	AU	missing data
Balkau et al. [40] 2010	OR: 9.33 (5.05–17.25) for males OR: 36.72 (17.12–78.76) for women	missing data	missing data	<20 ≥70	NAFLD-LFS	*p* < 0.0001: FLI vs. NAFLD-LFS
Zelber-Sagie et al. [41] 2013	0.65 FLI vs. AU 0.97 (0.95–0.98): FLI vs. SteatoTest 0.82 (0.77–0.87): FLI vs. HRI:	80.3 (FLI vs. AU) 85.5 (FLI vs. SteatoTest) 56.3 (FLI vs. HRI)	87.3 (FLI vs. AU) 92.6 (FLI vs. SteatoTest) 86.5 (FLI vs. HRI)	≥60	AU, SteatoTest and HRI	*p* < 0.001: FLI vs. SteatoTest *p* < 0.001: FLI vs. HRI
Fedchuk et al. [42] 2014	0.83 (0.71–0.91)	76	87	>60	LBP	*p* < 0.006: FLI vs. LBP (when is steatosis >33%)
Kahl et al. [43] 2014	0.72 (0.59–0.85) FLI 0.79 (0.68–0.9) HSI 0.7 (0.53–0.88) NAFLD-FLS	76 100 35	83 91 91	>60 >36	^1^H-MRS, HSI, NAFLD-FLS	*p* < 0.01: ^1^H-MRS *p* < 0.001: HSI *p* < 0.05: NAFLD-LFS
Goulart et al. [44] 2015	0.76 (0.69–0.83)	89.6 79.1	36.7 64.1	<30 ≥60	AU, CT	Missing data
Xia et al. [45] 2016	0.76(0.74–0.77: Chinese pts 0.72(0.66–0.77: Finnish pts	57 (53–60) in Chinese pts. 85 (79–90) in Finnish pts.	81 (79–82) Chinese pts. 45 (38–53) Finnish pts.	>27.1 in Chinese pts. >39 in Finnish pts.	AU	*p* < 0.001: FLI vs. AU Chinese and Finns
Motamed et al. [46] 2016	0.8656 (0.8548–0.8764)	82.42 82.33	76.87 76.55	>46.9: males >53.8: women	AU	*p* < 0.0001: FLI vs. AU
Dehnavi et al. [47] 2018	0.85 (0.79–0.9)	83	70	>26.2	CAP	*p* < 0.001: FLI vs. CAP
Lind et al. [48] 2020	0.82 (0.80–0.83)	missing data	missing data	missing data	MRI	*p* = 0.0019: LFS vs. FLI (in the EFFECT group) *p* = 0.005: FLI vs. LAP (in the POEM group)
Chen et al. [49] 2020	0.84 (0.81–0.87): general population 0.793 (0.748–0.839): males 0.765 (0718–0.813): women	80.3: males 76.1: women	66.9: males 65.5: women	>20: males >10: women	AU	*p* < 0.001: FLI vs. AU
Jung et al. [50] 2020	0.68 (0.64–0.71)	71.2 34.4	57.8 83.4	<30 ≥60	MRI	*p* < 0.01: FLI NAFLD and FLI no NAFLD
Zhang et al. [51] 2021	0.78 (0.71–0.86)	87	58.5	>37.64	MRI-PDFF	missing data
Castellana et al. [52] 2021	0.14 (0.09–0.19) 0.42 (0.34–0.51) 0.67(0.58–0.75)	81 (71–88) - 44 (33–55)	65 (52–76) - 90 (84–94)	<30 30–60 ≥60	AU vs. CT/MRI	missing data
Preciado-Puga et al. [53] 2021	0.704 (0.567–0.841)	71	51	<30 >60	AU and LBP	*p* = 0.011: FLI vs. AU *p* = 0.012: FLI vs. LBP
Borges-Canha et al. [54] 2022	0.31 (0.28–0.37) 0.22 (−0.81–1.24)	missing data	missing data	>30	FLI and BAARD score	*p* < 0.01: FLI vs. WC *p* < 0.01:FLI vs. WHR
Shao et al. [55] 2022	0.59 for NAFLD 0.83 for MAFLD 0.17 for non-MAFLD-NAFLD	missing data	missing data	>60	LBP, AU, MRI-PDFF, and CAP	missing data
de Silva et al. [56] 2022	0.692 (0.565–0.786)	58.33	69.49	>30: children (5–15 years)	AU	*p* = 0.0001: FLI vs. AU
Ali et al. [57] 2022	0.999 (0.997–1.00): FLI	98 83	100 100	<30 >60	AU, CAP	*p* = 0.0001: FLI vs. HSI *p* = 0.001: FLI vs. ZJU
Kim et al. [58] 2023	0.813 for mild steatosis	85	77	>30 ≥60	MRI-PDFF	*p* < 0.001
Mertens et al. [59] 2023	0.83 (0.74–0.91)	62	68	>60	AU, FLI, CAP and ^1^H-MRS	AU vs. CAP vs. FLI (*p* < 0.001) FLI vs. CAP (*p* = 0.684)
Kamari et al. [60] 2023	0.83 (0.825–0.842)	missing data	missing data	>60	clinical and biological data	*p* < 0.001: FLI vs. WC *p* < 0.001: FLI vs. TGs

Abbreviations: Ref: reference; AUC: area under curve; 95% CI = 95% confidence interval; NAFLD: non-alcoholic fatty liver disease; FLI: fatty liver index; * *p* Value: the significance of the statistical test that analysed the concordance between FLI and other diagnostic methods; ≥: presence of steatosis; AU: abdominal ultrasound; <: absence of steatosis; NAFLD-LFS: NAFLD liver fat score; HRI: hepatorenal ultrasound index; LBP: liver biopsy puncture; ^1^H-MRS: H-magnetic resonance spectroscopy; CT: abdominal computed tomography; CAP: controlled attenuation parameter; MRI: magnetic resonance imaging; BAARD score: an algorithm that combines three variables: BMI, AST/ALT ratio, and the presence of diabetes into a weighted sum (BMI > 28 = 1 point, AST/ALT ratio of >0.8 = 2 points, diabetes = 1 point), to finally generate a score from 0 to 4; WC: waist circumference; WHR: waist-to-hip ratio; ZJU: Zhejiang University index; MRI-PDFF: magnetic resonance imaging derived proton density fat fraction; TGs: triglycerides.

**Table 2 diagnostics-13-03316-t002:** The accuracy and limit values of FLI in NAFLD screening in people and patients who presented risk factors, in the most representative studies in the literature.

Author [Ref], Year	AUC (95% CI) FLI	Sensitivity %	Specificity %	Cut-Off FLI	Reference Methods for NAFLD Diagnosis	* *p* Value
Nishi et al. [63] 2015	1.18–4.70: for males 1.07–8.19: for women	missing data	missing data	<30 ≥60	FLI	*p* = 0.003: for males *p* < 0.001: for women
Li et al. [64] 2018	0.74 (0.67–0.81): weak population 0.83 (0.80–0.86): total population 0.71 (0.66–0.77): overweight/obese population	84.2	65.3	>20	AU	*p* < 0.01
Klisic et al. [65] 2018	0.909 for model 2	89.3: model 2 vs. model 1	87.5: model 2 vs. model 1	<30 ≥60	FLI	*p* < 0.001: BMI and WC were statistically higher in the two FLI groups
Chen et al. [12] 2019	0.802 (0.762–0.839)	66	80	<30 ≥60	AU, FLI and HSI	*p* = 0.0383: FLI vs. HSI
Khang et al. [66] 2019	0.849 (0.841–0.856)	0.280	0.035	<20 ≥60	FLI and AU	*p* < 0.001
Hsu et al. [67] 2019	0.76 (0.73–0.78	60.66	79.35	≥60	AU	*p* < 0.001
Motamed et al. [68] 2020	0.712 (0.675–0.749): for males 0.721 (0.683–0.759): form women	missing data	missing data	missing data	AU	*p* < 0.001
Busquets-Cortes et al. [69] 2021	6.343 (5.368–7.495)	missing data	missing data	<30 ≥60	FLI	*p* < 0.001

Abbreviations: Ref: reference; AUC: area under curve; 95% CI = 95% confidence interval; NAFLD: non-alcoholic fatty liver disease; FLI: fatty liver index; * *p* Value: the significance of the statistical test that analysed the concordance between FLI and other diagnostic methods; ≥: presence of steatosis; AU: abdominal ultrasound; <: absence of steatosis; BMI: body mass index; WC: waist circumference; HSI: hepatic steatosis index.

**Table 3 diagnostics-13-03316-t003:** The accuracy and limit values of FLI in monitoring the effects of different therapeutic interventions on NAFLD, in the most representative studies in the literature.

Author, Year [Ref]	AUC (95% CI) FLI	Sensitivity %	Specificity %	Cut-Off FLI	Reference Methods for NAFLD Diagnosis	* *p* Value
Kanerva et al. [81] 2014	0.68 (047–0.84)	61	86	>60	NAFLD-LFS	*p* < 0.001 NAFLD-LFS vs. FLI
Huh et al. [82] 2015	1.75 (1.39–2.20): FLI 1.39 (1.26–1.55): HSI	Missing data	Missing data	≥60 >36	HSI, BARD- score	*p* < 0.001: FLI vs. estimated 24 h sodium excretion
Cantero et al. [83] 2017	0.84	Missing data	Missing data	≥60	FLI, HSI and NAFLD-LFS	*p* = 0.793: RESMENA vs. AHA
Cantoral et al. [84] 2019	0.88 (0.74–1.06)	77.8	61	≥30	MRI	*p* < 0.01: FLI vs. HSI
Gastaldelli et al. [85] 2020	EXE + DAPA vs. EXE + placebo:−511, −0.73 EXE + DAPA vs. DAPA + placebo:−4.93, −0.62	Missing data	Missing data	≥60	FLI and NAFLD-LFS	*p* = 0.0092 *p* = 0.0119
Nucci et al. [86] 2023	0.73 (0.33–0.89): before diet 0.85 (0.54–0.95): after diet	Missing data	Missing data	>60	CAP, FAST score	*p* < 0.0001 for FLI

Abbreviations: Ref: reference; AUC: area under curve; 95% CI = 95% confidence interval; NAFLD: non-alcoholic fatty liver disease; FLI: fatty liver index; * *p* Value: the significance of the statistical test that analysed the concordance between FLI and other diagnostic methods; ≥: presence of steatosis; <: absence of steatosis; pts: patients;; NAFLD-LFS: NAFLD liver fat score;; HSI: hepatic steatosis index;; BARD score: the association of BMI, AST/ALT ratio, and presence of diabetes; RESMENA: Reduction of Metabolic Syndrome in Navarra; AHA: American Heart Association; CAP: controlled attenuation parameter; MRI: magnetic resonance imaging; DAPA: dapaglifozina; EXE: exenatida; FAST: FibroScan-AST.

## Data Availability

Not applicable.

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
