# Peer review of "The Role of the Fatty Liver Index (FLI) in the Management of Non-Alcoholic Fatty Liver Disease: A Systematic Review"

_diagnostics, 2023, doi:10.3390/diagnostics13213316_

Round 1
Reviewer 1 Report
A useful and interesting paper for practice.
Several suggestions:
Write a paragraph introducing the new fatty liver nomenclature, which is SLD, MASLD, and MASH!
Page 2, Row 99: Ultrasound is not as expensive as other imaging modalities!
It is not very valid to compare FLI with standard abdominal US because it depends on the amount of fatty liver.
For the quantification of fatty liver, a CT scan is advised everywhere (radiation risk)
Perhaps some advice on when to use other biological tests and when to use FLI.
Lastly, avoid saying that FLI will be utilized for monitoring the effects of various therapies in your conclusion.
Author Response
Dear Reviewer!
Thank you very much for taking the time to review this article. I particularly appreciate your effort and the very welcome advice for improving the quality of the presented material.
Next, I will detail the changes made to our article in accordance with your suggestions.
1. Write a paragraph introducing the new fatty liver nomenclature, which is SLD, MASLD, and MASH!
In the chapter ”Material and Methods”, Part I, we introduced the definitions of SLD, MASLD, and MASH.
- Page 2, Row 99: Ultrasound is not as expensive as other imaging modalities!
Thank you for your attention. I excluded the term ultrasonography and left only CAP and MRI.
- It is not very valid to compare FLI with standard abdominal US because it depends on the amount of fatty liver.
In lines 12-14 we mentioned that the results obtained by calculating FLI, HSI, and SteatoTset will be validated by echography, so abdominal ultrasound is used for the diagnosis of HS, while calculating FLI is only necessary for the selection of patients in whom it is necessary to we confirm HS.
- For the quantification of fatty liver, a CT scan is advised everywhere (radiation risk)
Indeed, the use of CT for the diagnosis of HS should be greatly restricted due to the irradiating effect of this method. I mentioned this fact in the text of the manuscript.
- Perhaps some advice on when to use other biological tests and when to use FLI.
Throughout the manuscript, we tried to give this useful information to the readers.
Biochemical tests are part of the initial investigation of all patients with liver symptoms, and FLI calculation is especially indicated in those patients who present biochemical changes suggestive of NAFLD.
Afterward, there is the diagnostic stage of NAFLD, which requires the performance of imaging tests and liver biopsy.
- Lastly, avoid saying that ”FLI will be used for monitoring the effects of various therapies” in your conclusion.
We have replaced this statement, saying that in the future "the FLI will be an easy-to-use tool in evaluating the benefits of these therapies."
Kind regards,
Dr Biciusca Viorel.
Reviewer 2 Report
Abstract. Do not use acronyms in this section since the introduction are also present.
Introduction
line 40 NAFLD, especially evolving in non-alcoholic steatohepatitis (NASH).
lines 57-59 the blood biomarkers introduced are liver transaminase levels and triglycerides. The FLI and HSI aren’t biomarkers per se but algorithms. FLI is an algorithm combining body mass index (BMI), waist circumference (WC), gamma-glutamyl transferase (GGT), and triglyceride (TG) levels. Either, HSI is an algorithm combining gender, BMI, waist circumference (WC), and transferases (AST and ALT) levels and the presence or not of type 2 diabetes, as you stand. So please use a sentence like: the blood biomarkers introduced are liver transaminase levels and triglycerides which are integrated into an algorithm FLI and HIS in order to have two biomonitoring tools.
Still at line 98 FLI, HSI, Steatotest, FIB-4 are indicated as serum biomarkers. Along the text, please refer to them as biomonitoring tools even in discussion and conclusions.
Since the work considers only data from humans, registering the SR in PROSPERO can be possible.
Make discussion easier and clear to read.
Author Response
Dear Reviewer,
Thank you very much for taking the time to review this article. I particularly appreciate your effort and the very welcome advice for improving the quality of the presented material.
Starting from your comments and suggestions, we transcribed the Material and Methods chapter. We mentioned that the study went through two stages. The first stage refers to "Diagnostic management of NAFLD". We researched the selected articles from the databases, about "NAFLD diagnosis and screening" in patients with metabolic risk. The second stage consisted of the research of articles related to "Therapeutic management of NAFLD". For each stage, we created a chapter on” Material and methods”, followed by Results.
At the end of the two stages, I rewrote the ”Discussions” chapter.
In addition, we made the specific changes you mentioned after reviewing the article.
1. Abstract. Do not use acronyms in this section since the introduction are also present.
Thank you for your attention. We gave up on acronyms in the abstract.
2. Introduction: line 40: NAFLD, especially evolving in non-alcoholic steatohepatitis (NASH).
You're right. We corrected! Thank you!
3. lines 57-59: the blood biomarkers introduced are liver transaminase levels and triglycerides.
The FLI inludes triglicerides and gamma-glutamyl transferase.
4. The FLI and HSI aren’t biomarkers per se but algorithms.
FLI is an algorithm combining body mass index (BMI), waist circumference (WC), gamma-glutamyl transferase (GGT), and triglyceride (TG) levels. Either, HSI is an algorithm combining gender, BMI, waist circumference (WC), and transferases (AST and ALT) levels and the presence or not of type 2 diabetes, as you stand. So please use a sentence like: the blood biomarkers introduced are liver transaminase levels and triglycerides which are integrated into an algorithm FLI and HIS in order to have two biomonitoring tools.
Thank you very much for your attention! It was careless of me. Throughout the manuscript, I have corrected the term biomarker with that of algorithm or tool when it refers to the term FLI or HSI:
E.g. lines 27-28: ...biomarkers panels;
line 33:...important algoritm;
line 63: ...biomarkers and non-invasive algorithms;
line 65: …non-invasive scores [14,15];
line 91: …serum biomonitoring tools;
line 98: I reformulated the whole sentence as follows: "Currently, the most effective screening methods for NAFLD and fibrosis (FLI, HSI, Steatotest, FIB-4) are those based on serum biomarkers as well as imaging methods such as transient elastography and MRI, although their cost is quite high”
line 108: …blood biomarkers panels;
line 116; … as many non-invasive tools
line 220:… with other diagnostic biomarkers and scores
line 221: … other non-invasive biomonitoring tools or scores
line 304: … investigated methods for HS assessment
line 517: …of these non-invasive tools has limited
line 522: ,,, of non-invasive scores
line 523: … non-invasive index used in
line 598: The other three scores currently used
line 606: … among the other biomonitoring tools
5. Still at line 98 FLI, HSI, Steatotest, and FIB-4 are indicated as serum biomarkers.
line 98: We reformulated the whole sentence as follows: "Currently, the most effective screening methods for NAFLD and fibrosis (FLI, HSI, Steatotest, FIB-4) are those based on serum biomarkers as well as imaging methods such as transient elastography and MRI, although their cost is quite high"
- Along the text, please refer to them as biomonitoring tools even in discussion and conclusions.
Thank you very much! I replaced in all the positions where I used the term "biomarker" incorrectly with terms synonymous with "biomonitoring tools".
7. Since the work considers only data from humans, registering the SR in PROSPERO can be possible.
I introduced a sub-chapter on Limitations, where I explained the reason why I did not register the PRISMA report in the PROSPERO public platform.
”On the other hand, there are some limitations. First of all, considering the invasiveness and the risks to which the patients are subjected, liver biopsy, the gold standard for the diagnosis of hepatic steatosis, was not used as a mandatory diagnostic indicator in the selected studies. Secondly, considering the long waiting time for registration in the PROSPERO platform, the study protocol was no longer sent for registration in the public platform”.
- Make discussion easier and clear to read.
We transcribed the "Discussions" chapter so that the text here can be easier to read.
Kind regards.
Dr Biciusca Viorel.
Round 2
Reviewer 2 Report
Excellent revision, just two other things. 1) Cancel MD near the author's name, and 2) Since authors decided not to go for Prospero Registration, please try in the discussion section to see a new possibility in noncoding RNAs as potential biomarkers in the future; for reference, see 10.1016/j.biopha.2021.112132; 10.1016/j.cbi.2020.109199; 10.3390/cimb45010006.
Author Response
Dear Reviewer,
Thank you once again for your support during the manuscript evaluation process.
I believe that your suggestions were appropriate and contributed to increasing the quality of this article.
I hope that I managed to respond to the recommendations sent this time as well.
First, we deleted the MD acronyms from the authors of the manuscript.
Secondly, in the eighth paragraph of the Introduction, I replaced the phrase "Over the past 20 years" with the phrase "In the last two decades" written in green.
Thirdly, I completely replaced the tenth paragraph with a new phrase written in green. It is the paragraph that has reference number 32.
In addition to the Bibliography, reference number 32 has been completely replaced with another much more current reference (written in green).
Finally, please narrow paragraphs 7-11 from the Introduction into one so that all the theory related to fibrosis appears in one paragraph.
I wish you much success in your work!
Kind regards,
Dr Biciusca Viorel.